# Enhancement of the Anticancer Ability of Natural Killer Cells through Allogeneic Mitochondrial Transfer

**DOI:** 10.3390/cancers15123225

**Published:** 2023-06-17

**Authors:** Seong-Hoon Kim, Mi-Jin Kim, Mina Lim, Jihye Kim, Hyunmin Kim, Chang-Koo Yun, Yun-Joo Yoo, Youngjun Lee, Kyunghoon Min, Yong-Soo Choi

**Affiliations:** 1Department of Biotechnology, CHA University, Seongnam 13488, Republic of Korea; dkcnfm@chauniv.ac.kr (S.-H.K.); treasure7744@naver.com (M.-J.K.); nimmina@naver.com (M.L.); ckoo.yun@gmail.com (C.-K.Y.); yunjoo.yoo@gmail.com (Y.-J.Y.); 2Research & Development Division, Humancellbio Co., Ltd., Suwon 16227, Republic of Korea; yjlee@hcbio.co.kr; 3Department of Quantitative Health Sciences, Cleveland Clinic Lerner Research Institute, Cleveland, OH 44195, USA; 4Department of Genetics and Genome Sciences, Case Western Reserve University School of Medicine, Cleveland, OH 44106, USA; human.gim@gmail.com; 5Department of Rehabilitation Medicine, CHA Bundang Medical Center, CHA University School of Medicine, Seongnam 13496, Republic of Korea; minkhrm@gmail.com

**Keywords:** natural killer cells, immune cell therapy, mitochondria, mitochondrial transfer, anticancer immunotherapy

## Abstract

**Simple Summary:**

Conventional natural killer (NK)-based anticancer immunotherapy has a limitation: a culture period of approximately 2 weeks is required to increase the number and activity of NK cells. By transferring functional allogeneic mitochondria into NK cells, we demonstrated that the activity of NK cells and their cytotoxicity were significantly enhanced. This approach could potentially offer a timely therapeutic strategy for cancer treatment without the need for in vitro culture, which can be time-consuming and costly.

**Abstract:**

An in vitro culture period of at least 2 weeks is required to produce sufficient natural killer (NK) cells for immunotherapy, which are the key effectors in hematological malignancy treatment. Mitochondrial damage and fragmentation reduce the NK cell immune surveillance capacity. Thus, we hypothesized that the transfer of healthy mitochondria to NK cells could enhance their anticancer effects. Allogeneic healthy mitochondria isolated from WRL-68 cells were transferred to NK cells. We evaluated NK cells’ proliferative capacity, cell cycle, and cytotoxic capacity against various cancer cell types by analyzing specific lysis and the cytotoxic granules released. The relationship between the transferred allogenic mitochondrial residues and NK cell function was determined. After mitochondrial transfer, the NK cell proliferation rate was 1.2-fold higher than that of control cells. The mitochondria-treated NK cells secreted a 2.7-, 4.1-, and 5-fold higher amount of granzyme B, perforin, and IFN-γ, respectively, when co-cultured with K562 cells. The specific lysis of various solid cancer cells increased 1.3–1.6-fold. However, once allogeneic mitochondria were eliminated, the NK cell activity returned to the pre-mitochondrial transfer level. Mitochondria-enriched NK cells have the potential to be used as a novel solid cancer treatment agent, without the need for in vitro cytokine-induced culture.

## 1. Introduction

Natural killer (NK) cells are crucial in anticancer immunotherapy. The current trends of research on NK cell-mediated immunotherapy have focused on NK cell expansion, cytotoxicity, improved targeting, and lifespan extension [1]. To be applied clinically, NK cells need to be expanded via a mass culture for 2 weeks or more. However, in patients with terminal cancer, this activation induction period is insufficient for NK cell expansion. According to recent studies, the culture period can be shortened by culturing cells in a medium containing cytokines, such as IL-2 and IL-15 [2,3], or by using a formulation containing feeder cells and cytokines [4,5]. However, despite the induction of proliferation and activation, large-scale proliferation is limited, owing to the cost and cellular phenotype/functional differences [6]. Therefore, new strategies that can activate NK cell expansion are needed.

NK cell proliferation depends on the mitochondrial metabolic pathway [7,8]. Inactive NK cells mainly produce ATP through mitochondrial oxidative phosphorylation (OXPHOS) and have a relatively low basal metabolic rate [7,8]. However, when activated, glycolysis and OXPHOS increase ATP production [7,8]. Recent studies have shown that NK cell proliferation is inhibited when mitochondria-related signaling mechanisms are suppressed [9]. In addition, the metabolic performance of normal and cytokine-stimulated NK cells is different, with both of these showing distinct mitochondrial polarization patterns [10]. Therefore, mitochondrial remodeling of OXPHOS has been proposed to be an essential gatekeeper of NK cell function.

The decreased antitumor capacity of NK cells is associated with damaged mitochondria [10]. Tumor-infiltrating NK cells exhibit highly fragmented mitochondria and reduced cytotoxicity and proliferation [9]. Furthermore, NK cells from the tumor site of patients with liver cancer have small and fragmented mitochondria in their cytoplasm, whereas those from healthy individuals and the peripheral NK cells outside the tumor have normal, large, and tubular mitochondria [11]. Mitochondrial fragmentation is correlated with reduced cytotoxicity and NK cell loss that leads to tumor evasion and a decreased NK cell-mediated tumor surveillance, reducing the survival rates of patients with liver cancer [10,11]. By regulating mitochondrial activity and effector functions, recent studies have demonstrated a reduction in cell cytotoxicity and cytokine production in PGC-1α-deficient-NK cells compared to those in naïve NK cells [12]. In addition, the transcriptional regulation of mitochondria mediated by PGC-1α plays an important role in anticancer immunity [12]. This evidence suggests that the mitochondrial function in NK cells plays a key role in innate immunity.

Recently, the transfer of healthy exogenous mitochondria to target cells has emerged as an attractive therapeutic strategy to treat damaged mitochondria. Studies have shown that mitochondria isolated from various sources can be transferred to damaged cells or tissues to restore the loss of function [13,14,15,16,17]. A recent study on immune cells demonstrated the activation of T cells and the promotion of regulatory T cell differentiation through the mesenchymal stem cell-mediated transfer of mitochondria [18]. However, to the best of our knowledge, no study to date has verified the efficacy of the direct transfer of mitochondria into NK cells. Therefore, in this study, we aimed to investigate whether the transfer of healthy exogenous mitochondria into NK cells affected their activity, in particular, NK cell expansion and anticancer regulatory ability. To enhance the mitochondrial capacity, mitochondria were isolated from normal hepatic cells (WRL-68) derived from liver tissue with high energy metabolism activity. The isolated mitochondria were transferred into NK cells using a simple centrifugation method described in a previous study [19]. After the transfer, we evaluated the proliferative and cytotoxic capacities of NK cells and the changes in the NK cell surface receptors. We demonstrated that mitochondria-enriched NK cells, which do not require in vitro cytokine-induced culture, have the potential to be used as a novel solid cancer treatment agent.

## 2. Materials and Methods

### 2.1. Cell Culture

This in vitro study was approved by the Institutional Review Board (IRB) of CHA University (Seongnam, Republic of Korea; IRB No.202204-BR-018-02). For the cell culture experiments, NK-92MI and WRL-68 cells were purchased from the American Type Culture Collection (Manassas, VA, USA). K562, MCF-7, HCT-116, OVCAR-3, and A549 cells were purchased from the Korean Cell Line Bank (Seoul, Republic of Korea). Primary NK cells were obtained from donors using an NK cell isolation kit (Miltenyi Biotec, Bergisch Gladbach, Germany) and the Ficoll method. The NK-92MI cells were cultured in Minimum Essential Medium Eagle Alpha Modification (HyClone Laboratories Inc., Logan, UT, USA) supplemented with 12.5% fetal bovine serum (FBS; Gibco, Waltham, MA, USA), 12.5% horse serum (Gibco), 0.1 mM 2-mercaptoethanol (Gibco), 0.2 mM myo-inositol (Sigma-Aldrich, St. Louis, MO, USA), and 0.02 mM folic acid (Sigma-Aldrich). The WRL-68 cells were cultured in Dulbecco’s Eagle’s minimal essential medium (HyClone) supplemented with 10% FBS. The K562, MCF-7, HCT-116, OVCAR-3, and A549 cells were cultured in the RPMI-1640 medium (HyClone) supplemented with 10% FBS. Primary NK cells were cultured in ALys505NK (CSTI, Sendai, Japan) supplemented with 10% FBS, 200 IU/mL IL-2, 5 ng/mL IL-12, and 50 ng/mL IL-18. All cells were cultured at 37 °C under 5% CO_2_.

### 2.2. Isolation of the Mitochondria

Mitochondria were isolated using a method described in a previous study, with slight modifications [19]. Briefly, the WRL-68 cells (1 × 10^7^) were homogenized in the SHE buffer (0.25 M sucrose, 20 mM HEPES, 2 mM EGTA, 10 mM KCl, 1.5 mM MgCl_2_, and 0.1% defatted bovine serum albumin; pH 7.4) using a 1 mL disposable syringe and centrifuged at 1500× *g* for 5 min. To obtain the mitochondria, the supernatant was centrifuged at 20,000× *g* for 10 min. The mitochondrial pellet was resuspended in 1.8 mL of the SHE buffer and centrifuged at 20,000× *g* for 5 min. Then, the pellet was resuspended in 1.8 mL of the respiration buffer and centrifuged at 20,000× *g* for 5 min to obtain pure mitochondria. All procedures were carried out on ice, and the freshly isolated mitochondria were stored at 4 °C.

### 2.3. Quality Control of the Mitochondria

The mitochondrial content was measured using the Pierce bicinchoninic acid protein assay kit (Thermo Scientific, Waltham, MA, USA). The ATP content of the isolated mitochondria was measured using the CellTiter-Glo 2.0 (Promega, Madison, WI, USA) reagent, following the manufacturer’s protocol. To assess its purity, the isolated mitochondria were labeled with MitoTracker Green (Invitrogen, Waltham, MA, USA) and APC-conjugated TOMM20 (Abcam, Cambridge, UK; ab225341), and were analyzed using the CytoFLEX flow cytometry (Beckman Coulter, CA, USA). Isolated mitochondria were characterized using Western blot analysis.

### 2.4. Transfer of Mitochondria to the Target Cells

The isolated mitochondria were transferred to the target cells, according to a method described in a previous study [19]. Briefly, recipient cells were added to a tube containing 100 µL of phosphate-buffered saline (PBS). The number of mitochondria refers to the mass of mitochondria (µg of protein) per 1 × 10^5^ recipient cells. The mitochondrial suspension in 10 µL of PBS was added slowly to each tube containing the recipient cells, and the tubes were then centrifuged at 1500× *g* for 15 min. All the tubes were stored at 4 °C.

After mitochondrial transfer, the NK-92MI cells were seeded at a density of 1 × 10^5^ cells/mL and cultured at 37 °C under 5% CO_2_. The NK-92MI cells were sub-cultured every 48 h. To track the intracellular retention period after the mitochondrial transfer, the NK-92MI cell mitochondrial DNA (mtDNA) expression, proliferative capacity, and cytotoxicity against K562 cells were analyzed at each passage point.

### 2.5. Cell Proliferation Assay

To determine the changes in cell proliferation after the mitochondrial transfer, the NK-92MI cells were seeded in 48- and 96-well plates at a density of 1 × 10^5^ cells/mL and incubated at 37 °C. Cell proliferation in each group was assessed using the WST-based EZ Cyto-X assay (DoGenbio, Seoul, Republic of Korea) every 24 h. At each time point, the WST solution was added to each well (final dilution: 1:10), followed by incubation for 2 h. The optical density at 450 nm (OD 450 nm) was measured using a Synergy HTX microplate reader (BioTek, Winooski, VT, USA). The WST-based EZ Cyto-X assay is a colorimetric method widely used to measure cell proliferation and viability. This assay was employed in our study to evaluate the impact of the mitochondrial transfer on cell proliferation.

### 2.6. Flow Cytometry Analysis

To validate the quantification results, the mitochondria isolated using MitoTracker Red CMXRos were transferred into the NK-92MI cells. The quantification of mitochondria was performed using the CytoFLEX flow cytometry. The mean fluorescence intensity (MFI) values were recorded in the PE channel, and regression analysis was performed using SigmaPlot 10.0 (Systat Software, San Jose, CA, USA).

After mitochondrial transfer, we analyzed cell viability, Ki-67, and cell cycle assays using flow cytometry. For the cell viability assay, the NK-92MI cells were cultured at 37 °C for 48 h after mitochondrial transfer. Then, they were resuspended in PBS containing 10 µg/mL propidium iodide (PI; Sigma-Aldrich, St. Louis, MO, USA) and analyzed using the CytExpert software, version 2.4. For the Ki-67 intracellular staining, the NK-92MI cells were fixed/permeated using a fixation/permeabilization kit (BD Biosciences, San Jose, CA, USA), following the manufacturer’s protocol. The NK-92MI cells were suspended in perm/wash buffer (BD Biosciences, Franklin Lakes, NJ, USA) with PE-conjugated Ki-67 (BioLegend; 350504, San Diego, CA, USA), and analyzed using the CytExpert software. For the cell cycle assay, the NK-92MI cells were analyzed 48 h after mitochondrial transfer. They were permeabilized in 70% ethanol at 4 °C for 30 min. The DNA was stained with 50 µg/mL PI and 200 µg/mL RNase A (iNtRON Biotechnology, Seoul, Republic of Korea) at 37 °C for 45 min. The cells were analyzed using the CytExpert software, and data analysis was performed using FlowJo 10.7.1 (BD Biosciences).

To analyze their cytotoxicity against cancer cells, the specific lysis ability of NK cells was analyzed using flow cytometry. Target tumor cells (K562 cells) were labeled with carboxyfluorescein diacetate succinimidyl ester (CFSE; Invitrogen). Then, 48 h after mitochondrial transfer, NK cells and CFSE-labeled target cells were co-cultured for 4 h. Then, 10 µg/mL PI was added to the co-cultured samples to identify the dead cells, which were analyzed using the CytExpert software. The specific cell lysis percentage was calculated using Equation (1):(1)$$specific\;cell\;lysis\;(\%)=\frac{experimental\;release\;(\%)-spontaneous\;release\;(\%)}{1-spontaneous\;release\;(\%)}\times 100$$

For the CD107a degranulation assay, 48 h after mitochondria transfer, NK-92MI cells were co-cultured with the same number of K562 target cells for 2 h. All the cells were harvested, stained with APC-conjugated CD56 (Invitrogen; 17-0567-42) and PE-conjugated CD107a (Invitrogen; 12-1079-42) for 30 min, and analyzed using the CytExpert software.

For cell phenotype analysis, the NK cells were stained using the monoclonal antibodies against the following proteins: NKG2D (Invitrogen; 11-5878-42), NKp30 (BioLegend, San Diego, CA, USA; 325210), NKp44 (BioLegend; 325108), NKp46 (BD Biosciences; 557991), CD69 (Invitrogen; 12-0699-42), OX40 (BioLegend; 350004), DNAM-1 (BioLegend; 338306), CD158 (BioLegend; 339504), CD158a (Invitrogen; 12-1589-42), CD158b (BioLegend; 312614), CD158e (Miltenyi Biotec; 130-116-822), NKG2A (Beckman Coulter; IM3291U), TIM-3 (Invitrogen; 11-3109-42), LAG-3 (BioLegend; 369306), TIGIT (BD, Ashland, OR, USA; 747846), and PD-1 (BioLegend; 367406). Data analysis was performed using FlowJo 10.7.1 (BD Biosciences).

### 2.7. Determining Granzyme B, Perforin, and Cytokine Levels

The amounts of proteins released were determined using ELISA kits (for granzyme B, IFN-γ, and TNF-α; R&D Systems, Minneapolis, MN, USA, and for perforin; Invitrogen) following the manufacturer’s protocol. Briefly, 48 h after mitochondrial transfer, NK-92MI cells were seeded in 48-well plates (1 × 10^5^ cells/well) and co-cultured with 1 × 10^4^ K562 cells for 4 h. Then, the culture medium was collected, and the protein levels were determined.

### 2.8. PCR Analysis

To validate the quantification results, the total RNA from NK-92MI cells was isolated using an easy-spin total RNA extraction kit (iNtRON Biotechnology), and cDNA synthesis was performed using the Maxime RT premix kit (iNtRON Biotechnology) according to the manufacturer’s protocol. cDNA was added to a PCR reaction mix containing the PCR master mix solution (iNtRON Biotechnology) and primers (NK-92MI mtDNA: forward, 5′-TTAACTCCACCATTAGCACC-3′ and reverse, 5′-GAGGATGGTGGTCAAGGGA-3′; WRL-68 mtDNA: forward, 5′-TGCCAGCCACCATGAATATC-3′ and reverse, 5′-GGTGGGTAGGTTTGTTGA-3′; and *GAPDH*: forward, 5′-GGAAGGTGAAGGTCGGAG-3′ and reverse, 5′-GGCAACAATATCCACTTTACC-3′). Products were loaded onto a 1.5 % agarose gel, and after agarose gel electrophoresis, they were observed under ultraviolet light.

### 2.9. Fluorescence Analysis

To confirm the mitochondrial transfer, NK-92MI cells were labeled with MitoTracker Green, and the mitochondria of WRL-68 cells were labeled with MitoTracker Red CMXRos (Invitrogen). After transferring the mitochondria into NK-92MI cells by centrifugation, the cell nuclei were counterstained with 4′,6-diamidino-2-phenylindole (Sigma-Aldrich). Image analysis was performed using the Axiovert 200M fluorescence microscope (Carl Zeiss, Jena, Germany). Digital images were generated using the Axiovision SE64 Rel 4.9.1 software (Carl Zeiss).

### 2.10. Western Blot Analysis

The isolated mitochondria were heat treated at 95 °C for 7 min using the SDS-PAGE loading buffer (LPS solution, Daejeon, Republic of Korea). After separating the proteins by size using a 12% SDS-PAGE gel, they were transferred to PVDF membranes. The membranes were treated with 3% BSA for 1 h, in order to block non-specific interactions, and then incubated overnight at 4 °C with each of the following primary antibodies: anti-AIF (sc-13116), anti-cytochrome C (sc-13156), anti-PCNA (sc-56) (all from Santa Cruz Biotechnology Inc., Santa Cruz, CA, USA), anti-COX IV (#4844s), and anti-GAPDH (#2118s) (all from Cell Signaling Technology, Beverly, MA, USA). After washing with TBST, the membranes were incubated with horseradish peroxidase (HRP)-conjugated secondary antibodies, goat anti-mouse IgG (1:1000; Santa Cruz Biotechnology; sc-516102-CM), and goat anti-rabbit IgG (Santa Cruz; sc-2357). The expression of the target proteins was visualized using an enhanced chemiluminescence system (ECL component from Pierce Clarity and Western ECL Substrate; Bio-Rad Laboratories, Hercules, CA, USA) and LAS-4000 camera (Fuji Photo Film, Tokyo, Japan).

### 2.11. Statistical Analyses

All statistical analyses were performed using SigmaPlot 12.0 (Systat Software, San Jose, CA, USA). Significant differences between groups were evaluated using Student’s *t*-test. Data are expressed as the mean ± standard deviation, and statistical significance was defined as *p* < 0.05.

## 3. Results

### 3.1. Successful Mitochondrial Isolation and Transfer into NK-92MI Cells Confirmed by the In Vitro Expression of Mitochondrial Markers

First, we isolated mitochondria from normal human hepatocytes (WRL-68 cells). We selected these donor cells since their ATP content was 5.2-fold higher than that of NK-92MI cells (Appendix A). The protein concentration in the isolated mitochondria was 148 µg/1 × 10^7^ cells (Appendix A). To determine whether the function of the isolated mitochondria was maintained, we measured their ATP content. The results showed that the ATP content increased in proportion to the mass of mitochondria (10, 20, 30, and 40 µg), confirming their functionality (Appendix A). In addition, the identity and purity of the isolated mitochondria were evaluated using two methods: flow cytometry and Western blotting (Appendix A). Flow cytometry analysis using MitoTracker-FITC and TOMM20-APC showed that 94% of the mitochondria were double positive, indicating that 94% of them did not rupture during isolation (Appendix A). Following the Western blot assay results, mitochondrial markers (AIF, COX IV, and cytochrome C) were present, whereas the PCNA and GAPDH proteins were almost absent (Appendix A).

Exogenous mitochondria were transferred into the NK-92MI cells by centrifugation, as described in a previous study [19]. The mitochondria from NK-92MI and WRL-68 cells were stained with the MitoTracker Green and Red CMXRos fluorescent dyes to detect the endogenous (green) and exogenous (red) mitochondria, respectively (Appendix A). The mitochondria isolated from WRL-68 cells were transferred to NK-92MI cells, and the colocalization of the mitochondria from the two different sources was indicated using yellow dye (Appendix A). The mitochondrial transfer to NK-92MI cells was also demonstrated by the expression of exogenous mtDNA (WRL-68 mtDNA) (Appendix A). The PCR results showed that the number of mitochondria transferred into NK-92MI cells was dose-dependent (Appendix A). Furthermore, the flow cytometry analysis revealed that the uptake efficiency of mitochondria was also dose dependent (Appendix A). The linear regression results showed a relationship between the relative MFI values and the mitochondrial content (Appendix A).

Furthermore, the centrifugal force during the transfer did not significantly alter the NK-92MI cell viability, cytotoxicity, or proliferation (Appendix A). Collectively, the mitochondria isolation from WRL-68 cells was performed with almost no contamination by other localized proteins, and so the mitochondrial transfer to NK-92MI cells was confirmed.

### 3.2. Mitochondrial Transfer Increased the Proliferative Capacity of NK-92MI Cells

We investigated the proliferative capacity of the NK-92MI cells every 24 h, following the transfer of different amounts of mitochondria (0, 0.05, 0.5, and 5 µg of protein). The cells that received mitochondria reached a stationary phase earlier than the control cells, and a significant difference was observed after 2 days (Figure 1A). The WST assay was performed 48 h after mitochondrial transfer, and the results of each mitochondrial transfer group presented significant dose-dependent differences (*p* < 0.05, *n* = 9) compared to those of the control group (Figure 1B). In the 5 µg group, the dose-dependent difference was 1.2-fold higher than that in the control group (Figure 1B). We also confirmed that mitochondrial transfer did not result in a significant change in cell viability within 48 h of the transfer (Figure 1C). Then, we investigated the relationship between the increased cell proliferative capacity due to mitochondrial transfer and cell cycle progression. Compared to the cell cycle phase in the control cells, the NK-92MI cells that received mitochondria shifted from the G0/G1 phase to the S phase (Figure 1D). In addition, the Ki-67 intracellular staining results showed that NK-92MI cell proliferation increased after the mitochondrial transfer (Figure 1E,F). Collectively, these findings suggest that mitochondrial transfer regulates the proliferation of NK92-MI cells.

### 3.3. Mitochondrial Transfer Increased the Cytotoxic Ability of NK-92MI Cells

We evaluated the changes in the cytotoxic ability of the NK-92MI cells post mitochondrial transfer. Cells retained their cytotoxic ability 48 h after the mitochondrial transfer. Moreover, the cells in the mitochondrial treatment group showed significantly higher cytotoxicity against K562 cells than the control group cells, and this cytotoxicity was dose-dependent (Figure 2A,B). In addition, after mitochondrial transfer, NK-92MI cells showed strong CD107a degranulation activity in response to K562 cells (Figure 2C,D). Accordingly, in the 5 µg group, the secretion of granzyme B and perforin increased by 2.7- and 4.1-fold, respectively (Figure 2E,F). In the presence of K562 cells, NK-92MI cells showed a relatively weak response, releasing low amounts of TNF-α (Figure 2G), whereas the NK-92MI cells treated with mitochondria produced 5-fold higher levels of IFN-γ than the control group cells (Figure 2H). A significant change in the TNF-α level was only observed in the case of the transfer of 5 µg of mitochondria (Figure 2G). In addition, no significant difference in the cellular phenotype was observed 48 h after mitochondrial transfer (Figure 2I,J). Next, we investigated the direct killing activity of NK-92MI cells that received mitochondria against various solid tumors. The lysis activity of the mitochondrial treatment group cells was significantly higher (1.3- and 1.6-fold compared to OVCAR-3 and HCT-116 cells, respectively) than that of the control group cells (Figure 3). These groups of cells also showed different lysis rates for A549 and MCF-7 cells compared to those of the control group cells; however, this difference was not significant (Figure 3).

### 3.4. The Transferred Allogeneic Mitochondria and Their Effects Are Eliminated over Time

To confirm the persistence of their proliferative/cytotoxic capacity, the NK-92MI cells treated with mitochondria were sub-cultured for five passages every 48 h. The PCR results showed that the transferred exogenous mitochondria in the NK-92MI cells were eliminated after passage 3 (approximately 100–150 h) (Figure 4A). Notably, the proliferative and cytotoxic abilities of the NK-92MI cells treated with mitochondria were also significantly reduced after passage 3, and did not differ from those of the control group (Figure 4B,C). We also confirmed the stability of the transferred mitochondria in both cell types (NK-92MI and primary NK cells) by image analysis (Appendix A). After transferring green fluorescent mitochondria (Mito-GFP) into the target cells, we tracked them during the culture process. The results confirmed that the fluorescence was maintained for up to 8 days.

### 3.5. Transfer of Mitochondria into Primary NK Cells Enhances their Killing Activity

Finally, we investigated whether mitochondrial transfer is applicable to clinical studies. Primary NK cells isolated from human blood were confirmed to be cytotoxic 48 h after isolation (immediately, because they were not activated). Primary NK cells were incubated with exogenous mitochondria. The mitochondrial transfer into primary NK cells led to a higher killing activity against K562 cells compared to that of the control cells, as evidenced by their enhanced cytotoxic capacity (Figure 5).

## 4. Discussion

The immune surveillance capacity of NK cells is closely related to the number and activity of the endogenous mitochondria [20]. We demonstrated that transferring exogenous allogeneic mitochondria to NK cells can increase their immune activity. The transfer of exogenous mitochondria to NK cells promotes their proliferation and anticancer activity. Therefore, NK cells with an increased anticancer ability can be produced in a short period of time by mitochondrial transfer, without the need for a 2-week culture.

Recently, we demonstrated that impaired cellular energy metabolism can be repaired by the transfer of foreign mitochondria into cells [19]. After the transfer of healthy exogenous mitochondria, the proliferation of mtDNA-damaged ρ^0^ cells was restored [19]. Similarly, the proliferative capacity of NK cells treated with exogenous mitochondria was improved in a dose-dependent manner (Figure 1). In addition, as the proliferation rate of NK cells increased, the glucose consumption and lactic acid production in the medium increased rapidly. Therefore, we demonstrated that NK cell proliferation and ATP synthesis could be improved by exogenous mitochondrial transfer without the need for culture with cytokines.

The anticancer effect of NK-92MI cells increased in a dose-dependent manner after mitochondrial transfer (Figure 2). The cell anticancer effect did not increase immediately after mitochondrial transfer, but gradually increased over time. In addition, this effect differed to some extent depending on the type of target cell. However, the killing ability of NK-92MI cells treated with mitochondria proved to be effective not only against hematological cancer cells (K562) but also against solid tumor cells (Figure 3).

The problem with chimeric antigen receptor T-cell therapy, an anticancer immune cell therapy, is that side effects, such as an excessive cytokine storm, can persist for a long time in the body [20]. In addition, culturing or activating NK cells using cytokines is problematic because NK cells undergo apoptosis due to excessive stimulation. In the case of mitochondrial-enriched NK-92MI cells, no change in cell viability was observed, depending on the amount of mitochondria transferred (Appendix A), and the exogenous mitochondria were eliminated approximately 8 days after the transfer (Appendix A). Simultaneously, cell proliferation and the cell anticancer effects were restored to pre-mitochondrial transfer levels. As a result, mitochondria-enriched NK cells naturally returned to their original state after a week.

NK cell receptors play an important role in regulating NK cell activity [21]. When NK cells are cultured using cytokines, such as IL-2, IL-12, and IL-15, the expression of the activation receptors increases. The proliferation and activity of NK cells are enhanced by external signals. However, when exogenous mitochondria were transferred into cells, no changes in the activation receptors of the NK cells were observed (Figure 2). Therefore, the anticancer effect of mitochondria-enriched NK cells is due to an increase in the metabolic activity inside the cell, rather than an external signal. Conversely, if the number of mitochondria or their function decreases, the anticancer effect is also reduced.

As mentioned in previous studies, cell function enhancement is associated with mitochondrial states [22,23]. Although it is unclear how, the mitochondrial states may be improved via induction of mitochondrial biogenesis or fusion after mitochondrial transfer [24,25]. The artificial and natural transfer of mitochondria improves the function of normal and damaged cells [18,26,27,28]. These phenomena may explain the changes in cell metabolism associated with the mitochondria. First, studies have shown that mitochondrial transfer changes OXPHOS activity and ATP production [29,30]. The cytotoxicity and proliferation of NK cells involve the production of cytolytic granules and proliferative factors, respectively, which require a lot of ATP [8,31,32,33]. Our results suggest that an increased rate of ATP synthesis due to the transfer of mitochondria enhances NK cell functions, such as their killing activity, the release of cytokines and granules, and the cell cycle. Second, studies have reported that the changes in the mTOR signaling pathway mechanism induced by mitochondria are closely related to cell metabolism and the immune response [34,35,36]. In an mTOR inhibition model, the function and proliferation of NK cells decreased remarkably [35]. In addition, the hyperactivation of mTOR signaling leads to a decrease in NK cell function due to the mitochondrial fragmentation caused by autophagy [28]. Hence, we suggest that mitochondrial transfer might trigger an optimum regulation of the mTOR signaling pathway.

Gene set enrichment analysis (GSEA) using RNA-seq data also supports this observation at the transcriptome level (Appendix A). Based on these results, it can be concluded that the cytotoxicity and immune sensitivity of NK-92MI cells increased by upregulating gene expression in the Toll-like receptor and JAK-STAT signaling pathways (Appendix A). Interestingly, the observation that exogenous mitochondria upregulated glycolysis genes but downregulated OXPHOS genes suggests that there may be a balance between energy efficiency and productivity that is being optimized by the cell (Appendix A). One possibility is that the proportion of active versus inactive NK cells may play a role in this regulation. It is known that Glycolysis is more critical than OXPHOS for NK-receptor-activated cytotoxicity [8]. We observed that both pathways were upregulated at the protein level, but the transcriptomic level of OXPHOS was relatively downregulated. It is possible that the activation of NK cell function was acquired independently of the gene transcription involved in the OXPHOS pathway. This activation may be due to either sufficient protein abundance in OXPHOS, or other unknown mechanisms that altered the overall signal strength between protein and RNA levels in response to the proportion of active and inactive NK cell populations that already existed.

Further studies are needed to investigate whether the primary NK cells isolated from peripheral blood also exhibit anticancer activity against various solid cancer cells. Furthermore, whether mitochondria-enriched primary NK cells derived from peripheral blood naturally return to their original state needs to be confirmed in the future. Additionally, the mechanism by which the transferred mitochondria enhance the proliferation and killing ability of NK-92MI cells needs to be elucidated. Additionally, an important point to be considered is that the difference in cell anticancer effect according to the type of transferred mitochondria and the presentation of information on the donor of mitochondria should be supplemented.

## 5. Conclusions

In this study, we provide a concept at the in vitro level for the necessity of in vivo studies to accurately evaluate the toxicity of mitochondria-transferred NK cells to anticancer cells, and a general discussion of the results. Here, we demonstrated that the artificial transfer of allogenic mitochondria can improve the anticancer effect and proliferation of NK cells. When applying this strategy as an autologous NK cell therapy, we suggest transferring the mitochondria without additional cell culture. Moreover, we can combine this strategy with previous methods to expand and activate cells. Since this strategy leads to an increased cell proliferative capacity, the period of cell culture required to ensure a sufficient number of cells is produced can be shortened, and its associated cost can be lower. In addition, the results demonstrated that not only blood cancer cells, but also solid cancers, are affected by NK-92MI cells treated with mitochondria, suggesting that clinical trials are possible.

Furthermore, we demonstrated the functional changes in NK cells induced by exogenous mitochondrial transfer. The transfer of mitochondria with a relatively high ATP-producing ability into NK cells led to a mitochondrial function improvement. In addition, the mitochondrial transfer increased the proliferation of NK cells and enhanced their cytotoxicity. Collectively, mitochondrial transfer can be used to enhance NK cell activity in a short period and can be used as a novel therapeutic strategy for clinical application.

## Figures and Tables

**Figure 1 cancers-15-03225-f001:**
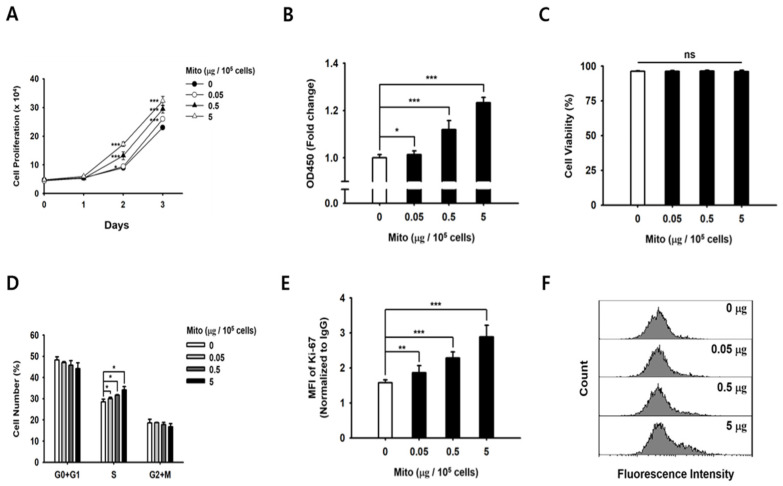
The NK-92MI cells that received mitochondria had an enhanced proliferative capacity. (**A**) Cell proliferation curve based on the data obtained every 24 h. (**B**) The WST assay results obtained 48 h after mitochondrial transfer. (**C**) Cell viability, 48 h after mitochondrial transfer. (**D**) Cell cycle analysis results, 48 h after mitochondrial transfer, presented as the % of cells in the G0/G1, S, and G2/M phases. (**E**) The mean fluorescence intensity of the Ki-67 intracellular staining of cells, 48 h after mitochondrial transfer. (**F**) Profile of FACS data of (**E**). Data are presented as the mean ± standard deviation, *n* = 9 (**A**–**C**,**E**–**F**) or *n* = 3 (**D**), * *p* < 0.05, ** *p* < 0.01, *** *p* < 0.001. ns, not significant.

**Figure 2 cancers-15-03225-f002:**
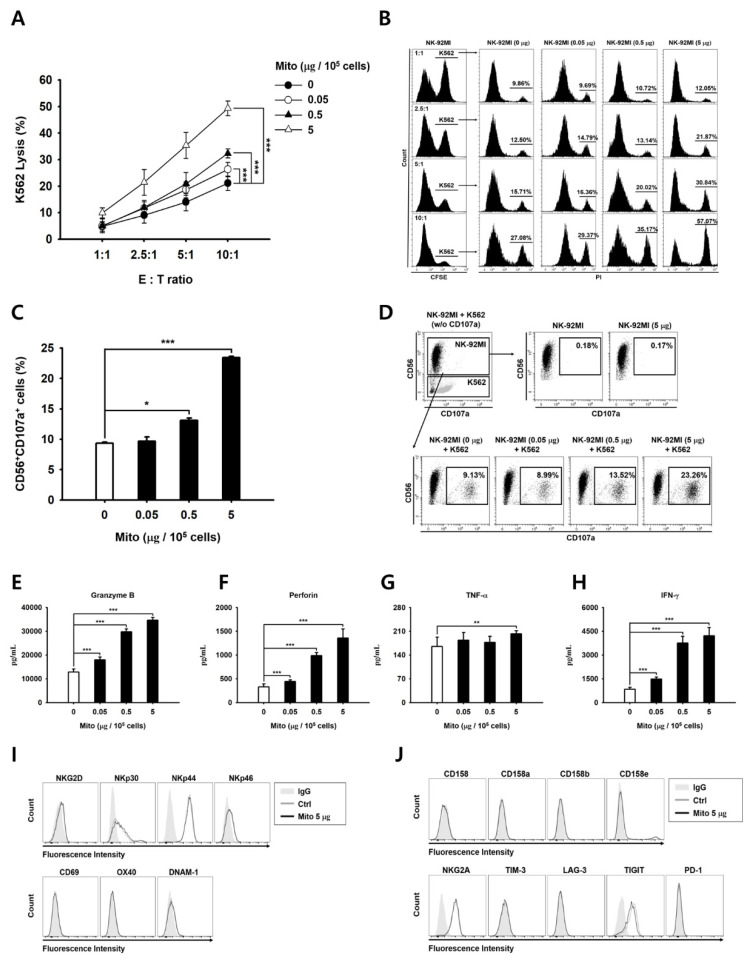
Mitochondrial transfer increased the in vitro cytotoxic capacity of NK-92MI cells against K562 cells. All the NK-92MI cells treated with mitochondria were cultured for 48 h. (**A**) Cytotoxicity assay was performed by analyzing the specific lysis of K562 cells after 4 h of co-incubation with NK-92MI cells. (**A**) Representative graph showing the lysis activity at several effector-to-target (E:T) ratios, 1:1, 2.5:1, 5:1, and 10:1, presented as the % of PI-positive cells. (**B**) Profile of FACS data of (**A**). (**C**) The CD107a degranulation assay was performed by incubating NK-92MI cells with the same number of K562 cells for 2 h; results are presented as the % of CD56-positive cells. (**D**) Profile of FACS data of (**C**). (**E**) Levels of granule granzyme B. (**F**) Perforin and cytokines ((**G**) TNF-α and (**H**) IFN-γ) in the supernatant of the NK-92MI cells co-cultured with K562 cells for 4 h. Phenotypic analysis of NK-92MI cells 48 h after mitochondrial transfer. (**I**) The levels of activating receptor and (**J**) inhibitory receptor were analyzed using flow cytometry. Data are presented as the mean ± standard deviation, *n* = 9 (**A**,**B**,**E**–**H**) or *n* = 3 (**C**,**D,I**,**J**), * *p* < 0.05, ** *p* < 0.01, *** *p* < 0.001.

**Figure 3 cancers-15-03225-f003:**
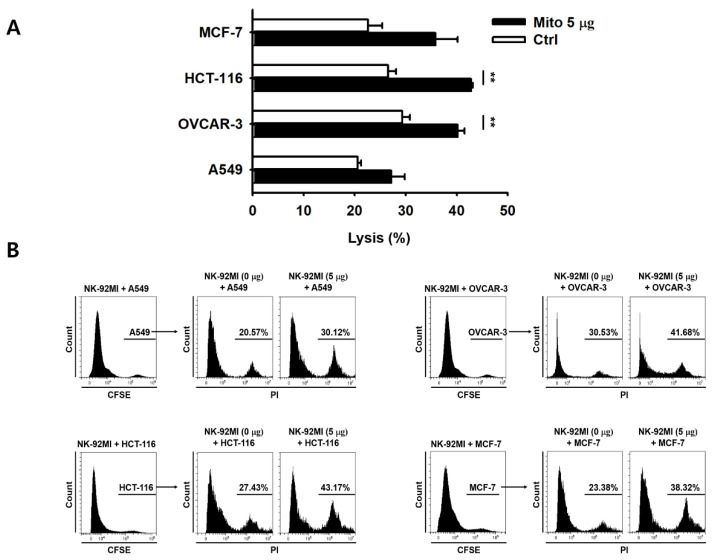
The killing activity of NK-92MI cells against various solid tumor cells was enhanced after mitochondrial transfer. Cells of adenocarcinoma of the breast (MCF-7), carcinoma of the colon (HCT-116), adenocarcinoma of the ovary (OVCAR-3), and carcinoma of the lung (A549) were used. (**A**) Representative bars showing the lysis activity against various solid tumor cells (T) after 4 h of co-incubation (E:T ratio = 10:1) with mitochondria-transferred NK-92MI cells (E), presented as the % of PI-positive cells. (**B**) Profile of FACS data of (**A**). Data are presented as the mean ± standard deviation, *n* = 3, ** *p* < 0.01.

**Figure 4 cancers-15-03225-f004:**
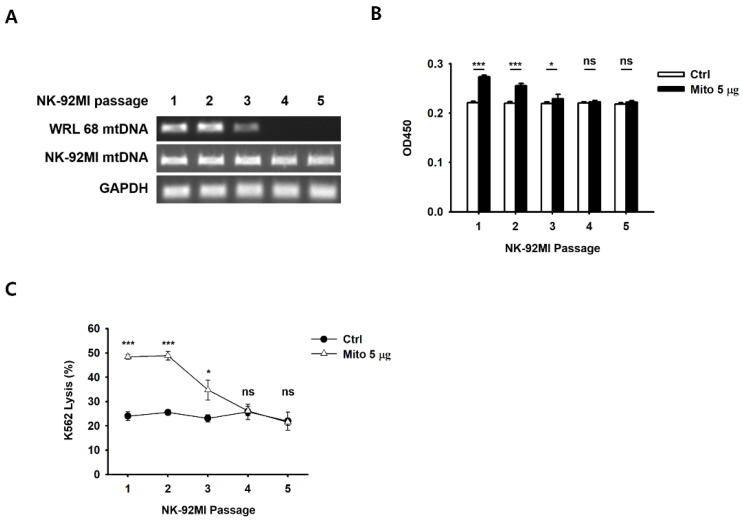
The transferred mitochondria (WRL-68 mtDNA) were eliminated from the recipient cells after a certain amount of time, as demonstrated by (**A**) the PCR results for the mitochondrial DNA obtained from the sub-cultures. NK-92MI cells were sub-cultured every 48 h, and reached an approximate confluence of 70–90%. (**B**) WST assay results demonstrating the cell proliferative capacity. (**C**) Specific lysis results demonstrating the cell cytotoxic capacity. Data are presented as the mean ± standard deviation, *n* = 9 (**B**) or *n* = 4 (**C**), * *p* < 0.05, *** *p* < 0.001. ns, not significant.

**Figure 5 cancers-15-03225-f005:**
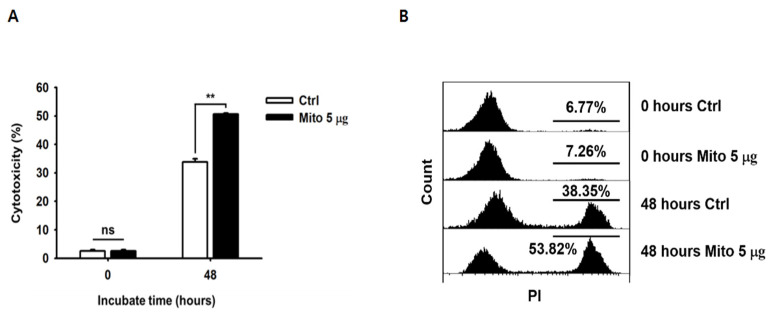
The in vitro killing activity of primary NK cells treated with mitochondria against tumor cells was enhanced. Representative bars showing specific lysis of K562 cell after co-incubation (E:T ratio =1:1) for 4 h with primary NK cells, presented as the % of PI-positive cells. (**B**) Profile of FACS data of (**A**). Data are presented as the mean ± standard deviation, *n* = 3, ** *p* < 0.01.

## Data Availability

The data presented in this study are available within the article or Appendix A.

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
