# Peer review of "Enhancement of the Anticancer Ability of Natural Killer Cells through Allogeneic Mitochondrial Transfer"

_cancers, 2023, doi:10.3390/cancers15123225_

Round 1

Reviewer 1 Report

In this study, the authors investigated whether the transfer of healthy exogenous mitochondria to NK cells affected their activity. The isolated healthy mitochondria were transferred into NK cells using a simple centrifugation method. After the transfer, the authors evaluated the NK cells, and identified that mitochondria-enriched NK cells, which do not require in vitro cytokine-induced culture, have the potential to be used as a novel solid cancer treatment agent. These results indicate a potential useful approach to manipulate NK cells for immunotherapy.

Main concerns:

1. The numbers of mitochondria should be quantitated after the mitochondria transfer in NK cells. The transferred NK cells will contain various numbers of mitochondria, which may impact the NK cell function.

2. After the healthy mitochondria transfer, the changes in the NK cell surface receptors should be presented in the result section. 

3. In vivo tumor treatments using mitochondria-transferred NK cells are needed for the overall statement of the research.

Minor concerns:

1. In figure 1E, in addition to the quantitation, presenting flow cytometric profiles are needed.

2. In figure 2B and figure 3, in addition to the quantitation, presenting flow cytometric profiles are needed.

3. The WST-based EZ-144 Cytox assay should be briefly introduced.

4. To increase the quality of the manuscript, profiles or images of flow cytometric analyses and fluorescence analyses should present.

Reviewer 2 Report

The authors Seong Hoon et al., gave an understanding of the enhancement of the anticancer ability of natural killer cells through allogeneic mitochondrial transfer.

The article is very interesting in the point of view that mitochondria transfer can be used as a novel therapeutic strategy for clinical application.

1. I suggest a second revision by an English Native of the manuscript in order to improve its quality. 

2. Please elaborate more on the conclusion to have a clear picture of the manuscript and the idea you want to be understood by the readers. I suggest adding an abstract scheme summarizing the general idea of the manuscript.

Reviewer 3 Report

In the manuscript titled “Enhancement of the anticancer ability of natural killer cells through allogeneic mitochondrial transfer”, authors performed serial assays to demonstrate that artificial transfer of allogenic mitochondria can enhance NK cells’ anticancer effect. This is an interesting manuscript providing a novel therapeutic strategy for cancer treatment. However, there are several concerns should be addressed.

1. In Figure 4, authors performed in vitro experiments showing that the transferred mitochondria were eliminated in a short period. How is the fate of mitochondria-enriched NK cells in vivo?

2. Mitochondria-enriched NK cells showed enhanced cytotoxicity against solid tumor cells in vitro. How is the effect of allogeneic mitochondrial transfer in tumor-bearing mice? Determining the effect of mitochondria-enriched NK cells is essential to prove the potential application of allogeneic mitochondrial transfer in NK cells.

Round 2

Reviewer 1 Report

The authors fairly addressed my previous concerns.

Author Response

Thank you so much for your detail and thoughtful comments on our work.

Reviewer 3 Report

NA

Author Response

As per reviewer's suggestion, we have performed an additional round of editing (3rd), which we believe has significantly improved the quality of the English language used in our submission. We have attached the certificate of editing for your reference.
